# Tumor Necrosis Factor and *Schistosoma mansoni* egg antigen omega-1 shape distinct aspects of the early egg-induced granulomatous response

**Kevin K. Takaki**[1], **Francisco J. Roca**[1], **Gabriele Schramm**[2], **Ruud H. P. Wilbers**[3], **Wannaporn Ittiprasert**[4,5], **Paul J. Brindley**[4,5], **Gabriel Rinaldi**[6], **Matthew Berriman**[6], **Lalita Ramakrishnan**[1]\*, **Antonio J. Pagán**[1]\*

**1** Molecular Immunity Unit, Department of Medicine, University of Cambridge, MRC Laboratory of Molecular Biology, Cambridge, United Kingdom, **2** Experimental Pneumology, Research Center Borstel, Airway Research Center North, Member of the German Center for Lung Research (DZL), Borstel, Germany, **3** Laboratory of Nematology, Plant Sciences Group, Wageningen University and Research, Wageningen, The Netherlands, **4** Department of Microbiology, Immunology and Tropical Medicine, School of Medicine and Health Sciences, George Washington University, Washington, DC, United States of America, **5** Research Center for Neglected Diseases of Poverty, School of Medicine and Health Sciences, George Washington University, Washington, DC, United States of America, **6** Wellcome Sanger Institute, Wellcome Genome Campus, Hinxton, United Kingdom

\* lr404@cam.ac.uk (LR); ap825@cam.ac.uk (AJP)

**Data Availability Statement:** All relevant data are within the manuscript and its Supporting Information files.

## Abstract

Infections by schistosomes result in granulomatous lesions around parasite eggs entrapped within the host tissues. The host and parasite determinants of the *Schistosoma mansoni* egg-induced granulomatous response are areas of active investigation. Some studies in mice implicate Tumor Necrosis Factor (TNF) produced in response to the infection whereas others fail to find a role for it. In addition, in the mouse model, the *S. mansoni* secreted egg antigen omega-1 is found to induce granulomas but the underlying mechanism remains unknown. We have recently developed the zebrafish larva as a model to study macrophage recruitment and granuloma formation in response to *Schistosoma mansoni* eggs. Here we use this model to investigate the mechanisms by which TNF and omega-1 shape the early granulomatous response. We find that TNF, specifically signaling through TNF receptor 1, is not required for macrophage recruitment to the egg and granuloma initiation but does mediate granuloma enlargement. In contrast, omega-1 mediates initial macrophage recruitment, with this chemotactic activity being dependent on its RNase activity. Our findings further the understanding of the role of these host- and parasite-derived factors and show that they impact distinct facets of the granulomatous response to the schistosome egg.

## Author summary

Schistosomiasis is a disease caused by parasitic flatworms which lay eggs within the veins of their human host. Upon sensing the parasite egg, macrophages, the first line defense

**Funding:** This work was supported by Wellcome Trust core-funding support to the Wellcome Sanger Institute (award number 206194) (GR, MB) and NIH MERIT award (R37 AI054503) and a Wellcome Trust Principal Research Fellowship (LR). These studies were additionally supported by the Wellcome Trust Strategic Award number 107475/Z/15/Z, and the NIAID Schistosomiasis Resource Center for distribution through BEI Resources, NIH-NIAID Contract HHSN272201000005I (PJB, WI). The funders had no role in study design, data collection and analysis, decision to publish, or preparation of the manuscript.

**Competing interests:** The authors have declared that no competing interests exist.

cells, aggregate tightly around the egg to encapsulate it within an immune structure known as a granuloma. These granulomas are the key pathological structures which determine both host disease outcome and parasite transmission. Studies in mice have implicated omega-1, a secreted parasite protein. Omega-1 is an RNase, an enzyme that degrades host RNA. Mouse studies have also suggested that a host defense protein, Tumor Necrosis Factor (TNF), is required to form granulomas around the egg. We used the small and transparent zebrafish larva to examine the requirement of omega-1 and TNF for granuloma formation. We find that omega-1 induces rapid macrophage migration and that its RNase activity is required for this. In contrast, TNF is not involved in the initial recruitment of macrophages. Rather, it enlarges granulomas after they are initiated. These findings improve our understanding of the role of omega-1 and TNF, and show that they impact distinct facets of granuloma formation around *Schistosoma* eggs.

## Introduction

Schistosomiasis is a major granulomatous disease, caused by parasitic flatworms of the genus *Schistosoma* with *Schistosoma mansoni* being the most widespread agent of the disease [1]. The events of *Schistosoma* egg-induced granulomas have been deduced mainly from histological assessments of human clinical samples and the use of experimental mammalian models [2,3]. We have recently reported the use of the optically transparent and genetically tractable zebrafish larva as a model to study early macrophage recruitment and granuloma formation in response to *S. mansoni* eggs [4]. Because the zebrafish larva lacks adaptive immunity during their first few weeks of development, this model can be used to dissect mechanisms in the sole context of innate immunity [4–6]. We found that while epithelioid granulomas form rapidly around mature eggs, immature eggs fail to provoke granulomas, consistent with the mature stage-specific secretion of antigens and their function to induce granuloma formation in mammalian models [4,7–12].

In the zebrafish, we can additionally examine macrophage recruitment within hours of implantation, and find that whereas injections of schistosome soluble egg antigen (SEA) obtained from mature eggs induce early macrophage recruitment, implantation of immature eggs fail to do so [4]. Together these findings both validate the zebrafish model to study *S. mansoni* egg-induced granuloma formation and reveal new insights into the underlying molecular mechanisms [4].

In mice, the cytokine Tumor Necrosis Factor (TNF) and the *S. mansoni* secreted antigen omega-1 have been identified as host and parasite factors, respectively, that promote granuloma formation around the egg [13–17]. However, the role of TNF remains controversial and the mechanism by which omega-1 exerts its role is unresolved. In this work, we use the zebrafish model to explore their roles in macrophage recruitment and innate granuloma formation.

## Materials and methods

### Ethics statement

All animal experiments were conducted in compliance with guidelines from the UK Home Office and approved by the Wellcome Sanger Institute (WSI) Animal Welfare and Ethical Review Body (AWERB).

### Zebrafish husbandry

All zebrafish lines were maintained on a recirculating aquaculture system with a 14 hour light—10 hour dark cycle. Fish were fed dry food and brine shrimp twice a day. Zebrafish embryos

were housed in fish water (reverse osmosis water containing 0.18 g/l Instant Ocean) at 28.5˚C. Embryos were maintained in 0.25 μg/ml methylene blue from collection to 1 day post-fertilization (dpf). At 24 hours post-fertilization 0.003% PTU (1-phenyl-2-thiourea, Sigma) was added to prevent pigmentation.

## Generation of the TNFR1 mutant and its usage

The zebrafish TNFR1 mutant (*tnfrsf1a*^rr19^) was generated using CRISPR Cas9 technology, targeting the sequence TGGTGGAAACAAGACTATGAA of the third exon of the gene (ENSG00000067182) using a T7 promoter-generated guide RNA. Sequencing verified the mutation as a 25 bp deletion (ATGAAGGGAAATTGTCTTGAAAATG) and 6 bp insertion (TGGTGG), resulting in a frame shift and introduction of a premature stop codon soon after the start codon. HRM genotyping was performed using the TNFR1-HRM1- forward and reverse primer set (5'-GTTCCCCACAGGTTCTAACCAG-3' and 5'-CTTGATGGCATTTAT CACAGCAGA-3', respectively). TNFR1 heterozygotes in the macrophage reporter background, *Tg(mpeg1:YFP)*^w200^ [18], were incrossed, genotyped, and sorted as fluorescence-positive, homozygous TNFR1 mutants or WT siblings. Homozygous TNFR1 mutants or WT siblings were then incrossed to generate larvae for experiments.

## Soluble egg antigens, WT and RNase mutant recombinant omega-1

For preparation of SEA, eggs were isolated from *S. mansoni*-infected hamsters as previously described [19], and then homogenized in PBS, pH 7.5, using a sterile glass homogenizer. The homogenate was then centrifuged at 21 krcf for 20 minutes. Supernatants were pooled and then dialyzed overnight in PBS using a 3.5 kDa molecular weight cutoff dialyzer. Sample was then centrifuged at 21 krcf for 20 minutes, and supernatant (SEA) was aliquoted and stored at -80˚C. SEA was quantified for protein concentration using the Micro-BCA assay (Pierce, 23225), and quality controlled by SDS-PAGE and western blotting against the *S. mansoni* antigens, omega-1, alpha-1, and kappa-5. Quality control for low LPS content was performed using the Chromo-LAL assay (Associate of Cape Cod, Inc., C0031-5). SEA from WT and corresponding omega-1 knockout eggs were injected at 1 ng per hindbrain ventricle. For comparison of SEA and plant-expressed omega-1, SEA was injected at 2 ng per hindbrain ventricle (1.5 nL injection of 1.4 mg/mL SEA), and plant-expressed omega-1 with LeX glycans [20] was injected at 0.02 ng per hindbrain ventricle, the relative concentration of omega-1 present in SEA (G. Schramm, personal communication). For DEPC inactivation of plant-expressed omega-1, 1 μL of 0.07 M DEPC (1/100 dilution of Sigma, D5758) was added to 5 μL of 1.5 mg/mL omega-1 (12 mM final concentration of DEPC), and then incubated for 1 hour at 37˚C. Because the small volume of protein did not allow for ultrafiltration and requantification of protein, the sample was simply diluted 1/100 in PBS and then 0.02 ng of protein injected into the hindbrain ventricle. For comparison, control sample was incubated at 37˚C (without DEPC-treatment) and then diluted 1/100 in PBS. Because the HEK-expressed WT and RNase mutant omega-1 (H58F) lack the native-like LeX glycans in plant-expressed and natural omega-1 [21,22], they were injected at a 5-fold higher concentration of 0.1 ng per hindbrain ventricle. All hindbrain injections of antigens were assayed at 6 hours post-injection.

## Hindbrain injection of antigens

Hindbrain injections were performed as previously described [5] using 2 ng of WT or Δω1 SEA, 0.02 ng of plant-expressed omega-1 untreated or DEPC-treated, or 0.1 ng of HEK-expressed WT or RNase mutant omega-1.

### Hindbrain implantation of eggs

Schistosome eggs were individually implanted into the zebrafish hindbrain ventricle as previously described [4]. Briefly, an incision was made into the zebrafish using a microinjection needle, after which an individual egg was passed though the incision and implanted into the hindbrain ventricle.

### Bacterial infections and quantification of infection burden

Bacterial infections and quantification of infection burden was performed as previously described [5]. Briefly, 75 CFU *Mycobacterium marinum* M strain was microinjected into the caudal vein of zebrafish larvae at 36 hours post-fertilization. At 4 days post-infection larvae were imaged by inverted fluorescence microscopy and bacterial fluorescence quantified from images.

### Confocal microscopy

Zebrafish were anesthetized in fish water containing tricaine and then and mounted onto optical bottom plates (MatTek Corporation, P06G-1.5-20-F) in 1% low melting point agarose (Invitrogen, 16520–100) as previously described [5]. Microscopy was performed using a Nikon A1 confocal laser scanning confocal microscopy with a 20x Plan Apo 0.75 NA objective and a Galvano scanner, acquiring 30–80 μm z-stacks with 2–3 μm z-step intervals. Timelapse microscopy was performed at physiological temperature using a heat chamber set to 28°C (Okolab) with an acquisition interval of 3 minutes.

### Granuloma quantification

Confocal images were used for quantifying the number of macrophages in contact with the egg, and subsequent classification of the immune response. Granuloma size was quantified by fluorescence analysis of confocal z-stacks which were flattened, and then fluorescent macrophages comprising the granuloma area was measured by fluorescent pixel counts (FPC) [5].

### Quantification of macrophage recruitment

Quantification of macrophage recruitment was performed by counting the number of fluorescently labeled macrophages within the hindbrain ventricle by fluorescence microscopy. Experiments quantifying macrophage recruitment following injection of egg antigens, utilized *Tg (mpeg1:Brainbow)^{w201}* larvae [23].

## Results

### TNF signaling through TNF Receptor 1 promotes macrophage recruitment to nascent *S. mansoni* egg-induced granulomas but is dispensable for initial macrophage recruitment to the eggs

The role of TNF in *S. mansoni* egg-induced granulomas remains unresolved after two decades of studies in the murine model of schistosomiasis. Early findings showed that *S. mansoni*-infected SCID mice were deficient in both granuloma formation and egg extrusion, phenotypes which was rescued by recombinant TNF and activated T cell medium, but not by TNF-depleted T cell medium [13]. These findings suggested a role for TNF in granuloma formation and egg excretion [13]. However, subsequent work from this group found that TNF knockout mice did not have a defect in granuloma formation [24]. Mice lacking both receptors through which TNF signals did exhibit a mild granuloma deficit, leading the authors to propose that it

might be due to a defect in signaling of the ligand lymphotoxin [24]. However, this would not explain their earlier findings that exogenous TNF rescued granuloma formation in SCID mice [13]. Meanwhile, a different group reported that TNF did not rescue granuloma formation in SCID mice [25]. Additionally, *S. mansoni*-infected SCID mice displayed normal levels of TNF expression, suggesting that other cells may be the major source of TNF during the infection [25]. It has been suggested that Ly6C$^{hi}$ monocytes, which are known to express TNF in response to the schistosome egg, might be the innate source of TNF [26].

To delineate the role of TNF in macrophage recruitment and granuloma formation around *S. mansoni* eggs, we used a TNFR1 zebrafish mutant created by CRISPR-Cas technology (see Materials and Methods). We first confirmed that the lack of TNFR1 signaling rendered zebrafish larvae susceptible to *Mycobacterium marinum* infection, consistent to our previous findings using TNFR1 morpholino [27] (**S1 Fig**). Next, we implanted the Hindbrain Ventricle (HBV) of wildtype and TNFR1 mutant larvae with *S. mansoni* eggs and evaluated granuloma formation after five days (Fig 1A–1C). We have recently categorized early macrophage recruitment and granuloma formation based on the number and characteristics of macrophages in contact with the egg: Minimal recruitment, 0–6 macrophages; Macrophages recruited, >6 macrophages; Granulomas, confluent epithelioid macrophage aggregates [4]. At 5 days post-implantation of the eggs, the TNFR1 mutants had similar macrophage responses to wildtype animals with ~50% of the animals forming epithelioid granulomas in each group (**Fig 1B**). However, we found that the TNFR1 deficient granulomas were significantly smaller than wildtype granulomas, with the mean granuloma size being 62% smaller than in wildtype (**Fig 1C and 1D**). We also noted that the TNFR1 mutant granulomas, though smaller, showed a characteristic epithelioid morphology with confluent macrophages and loss of intercellular boundaries. This finding suggest that epithelioid transformation may also be independent of TNFR1 signaling [4] **Fig 1D**). Because the *S. mansoni* granuloma is comprised solely of macrophages at this early stage [4], our findings imply that TNFR1 signaling would promote macrophage recruitment to a nascent granuloma around the egg. In the zebrafish model, we can also examine the initiation of macrophage recruitment to *S. mansoni* eggs within hours of implantation [4]. However, we found that TNFR1 signaling is not required for initiation of macrophage recruitment (**Fig 1E**).

Together, these results show that TNF signaling through TNFR1 is required specifically for macrophage recruitment after the initial macrophages reach the egg through other signal(s). Thereby, TNF mediates granuloma enlargement rather than granuloma initiation. Furthermore, TNFR1 is not required for epithelioid transformation. Finally, TNF plays a role in the granulomatous response in the sole context of innate immunity.

## *S. mansoni* omega-1 promotes initial macrophage recruitment to the egg through its RNase activity

Next, we wanted to probe the parasite determinants that induce granuloma formation. In recent work we found that immature *S. mansoni* eggs invoked neither granuloma formation nor even initial macrophage recruitment, indicating that mature egg antigens were required for the first macrophages to be recruited to the egg [4]. Mature eggs express a variety of antigens [7,28,29], of which omega-1 is known to be the major contributor to granuloma formation, as knockdown of its expression leads to greatly diminished granuloma formation around eggs [16,17]. Omega-1 is an RNase involved in several processes. In dendritic cells, it inhibits protein synthesis, alters cell morphology, induces IL-33 expression, and reduces conjugation affinity with T cells [21,22,30,31]. If and how this leads to granuloma formation is not known. However, it is well-established that its RNase activity is essential for inducing the Th2

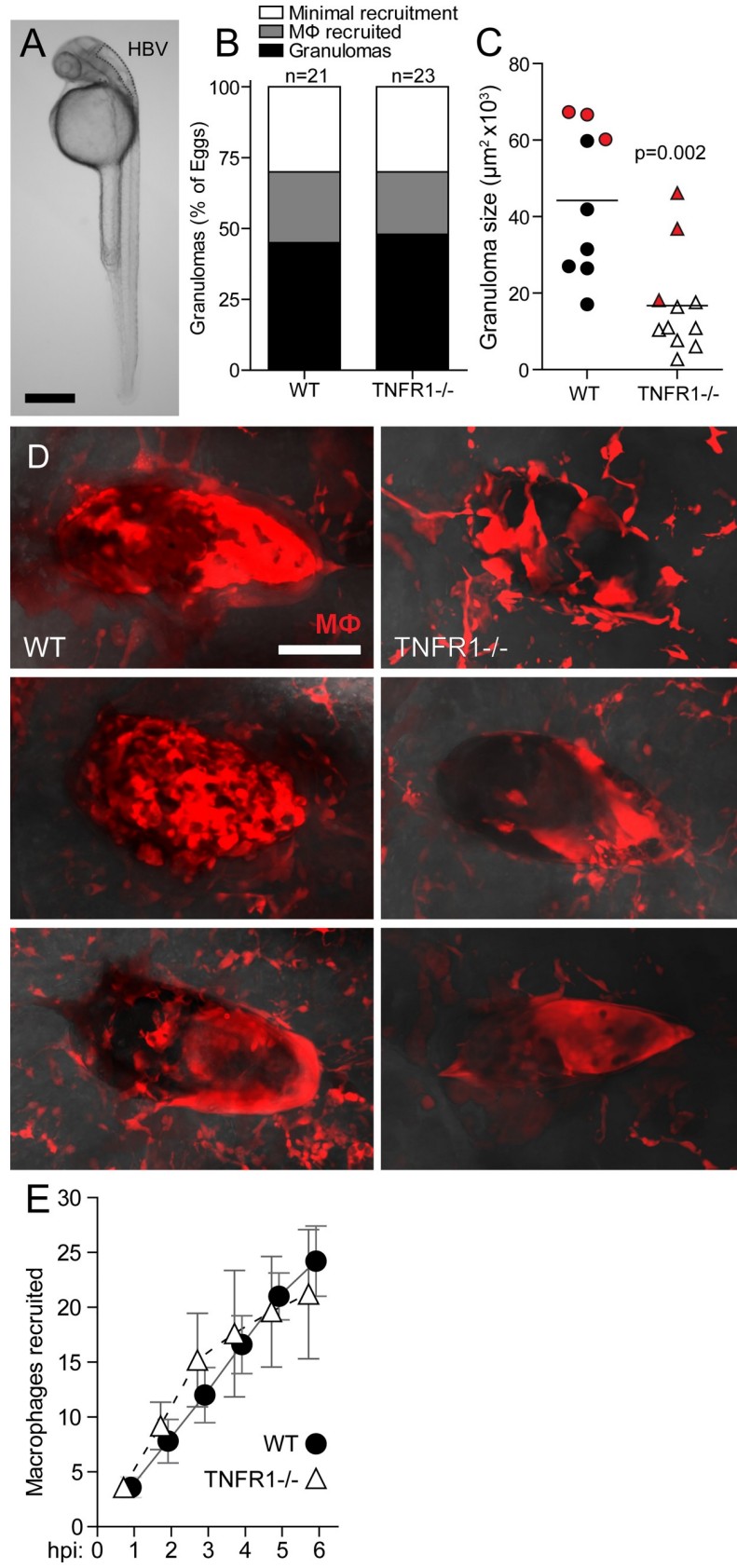

**Fig 1. TNF affects late-stage granuloma formation.** Comparison of macrophage recruitment and granuloma formation in WT and TNFR1 mutant zebrafish larvae with fluorescent macrophages following implantation with a

single schistosome egg into their hindbrain ventricle. (**A**) Zebrafish larva at 36 hours post-fertilization with the hindbrain ventricle (HBV) site of injection and implantation outlined. Scale bar, 300 μm. (**B-D**) Granuloma formation at 5 days post-implantation. (**B**) Percent of animals with; granuloma formation (confluent epithelioid macrophage aggregates), macrophages recruited (>6 macrophages in contact with the egg), or minimal recruitment (0–6 macrophages in contact with egg) [4]. (**C**) Granuloma size and (**D**) images, with each image from top to bottom corresponding with each red data point, top to bottom, respectively. Scale bar, 50 μm. Horizontal bars in (C), means. Statistics, Student's *t*-test. (**E**) Mean macrophage recruitment kinetics during the first 6 hours post-implantation. Error bars, SEM. Sample size, n = 5 animals per group.

polarization of granulomas [21,22,30,31]. This in turn induces expression of IL-4 and IL-13, known egg-induced host factors that can mediate granuloma formation [25,32,33]. Additionally, omega-1 is a major hepatotoxin [28,34,35], and it has been proposed that the granuloma itself would prevent the cytotoxic effects of this egg antigen on the host liver.

Our attempts to test the role of omega-1 by implanting omega-1 knockout (KO) eggs [17] into the larvae failed, as the genetically modified eggs did not survive shipment. As an alternative approach, we tested if the SEA obtained from omega-1 KO eggs could recruited macrophages. We examined macrophage recruitment 6 hours post-injection of the SEA into the hindbrain ventricle (**Fig 1A**). Omega-1-deficient SEA recruited macrophages similar to wild-type SEA (**Fig 2A**). Knockdown of omega-1 expression is not 100% efficient, with pools of CRISPR/Cas9-treated eggs still retaining ~20% of the omega-1 transcript, and their SEA still retaining ~20% of its RNase activity, suggesting that even though reduced compared to wild type eggs, the residual omega-1 may still be sufficient for macrophage recruitment [17]. Alternatively, the omega-1 activity may be redundant with other SEA components [36]. To investigate these hypotheses, we used a recombinant omega-1 that contain the native-like LeX glycosylation, which is important for its uptake by dendritic cells and subsequent Th2 polarization [20,21]. Injection of 0.02 ng of omega-1, the approximate amount of omega-1 in the corresponding SEA injections [4], induced macrophage recruitment, although less than SEA, consistent with other components inducing macrophage recruitment (**Fig 2B and 2C**).

Next, we asked if omega-1-associated recruitment of macrophages is dependent on its RNase activity. The inhibition of RNase activity in the recombinant omega-1 with diethyl pyrocarbonate (DEPC) [30], led to loss of macrophage-recruiting activity (**Fig 2D**). Because DEPC inhibits RNase function through covalent binding to the essential histidine in the

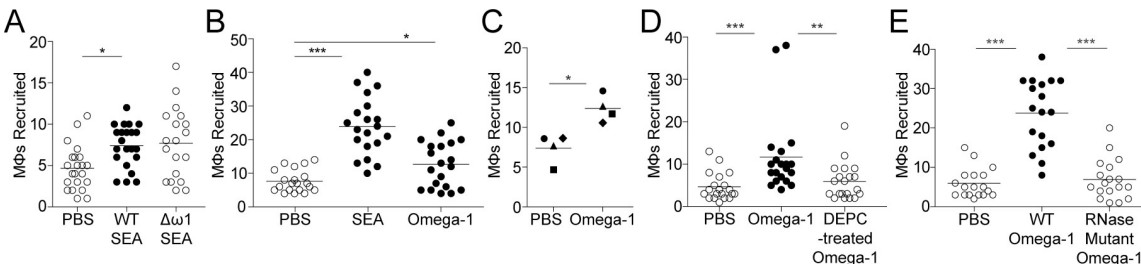

**Fig 2. Omega-1 recruits macrophages via its RNase activity.** Macrophage recruitment at 6 hours post-injection (hpi) with egg antigens. (**A**) Macrophages recruited to SEA from WT or omega-1 knockout eggs (Δω1). (**B**) Macrophages recruited to SEA or omega-1. (**C**) Mean macrophage recruitment to omega-1 for each of four experiments. Individual experiments represented with unique symbols; triangles and squares represent means of panels B and D, respectively. (**D**) Macrophages recruited to omega-1 or DEPC-treated omega-1. (**E**) Macrophages recruited to WT or RNase mutant omega-1. All omega-1 injections were performed using 0.02 ng of plant-expressed omega-1, with the exception of (**E**) which used HEK-expressed WT or mutant omega-1 injected at a 5-fold higher concentration of 0.1 ng to compensate for lack of LeX glycosylation in plant-expressed and natural omega-1. Statistics, ANOVA with Dunnett's post-test comparing all samples to PBS (**B**) or WT (**A,E**); (**D**) non-parametric ANOVA with Dunn's post-test comparing all samples to omega-1; (**C**) paired *t*-test. All horizontal bars, means. Statistics; * p<0.05, ** p<0.01, and *** p<0.001.

catalytic domain, one caveat is that it can create off-target modifications to the protein structure and function through binding to other histidine residues as well as, to a lesser extent, tyrosine, lysine, and cysteine [37]. To validate our findings, we used recombinant omega-1 mutant lacking RNase activity, with a phenylalanine substitution of the essential histidine of the catalytic domain [21,38]. As expected, the omega-1 mutant failed to recruit macrophages (**Fig 2E**). These findings confirmed that the omega-1 macrophage chemotactic activity is mediated through its RNase activity (**Fig 2B**).

## Discussion

This study reinforces the use of the zebrafish model to study molecular pathways involved in *S. mansoni* egg-induced granuloma formation. Particularly, it provides new insights on host and parasite factors modulating this critical process that drives the pathology associated with schistosomiasis.

We demonstrate that the cytokine TNF is required for granuloma enlargement but not initiation, in agreement with previous observations in the mouse [13–15,39]. Further, we show that TNF is dispensable for the first wave of macrophage recruitment to the egg. These findings are consistent with TNF not being a direct chemotactic agent, but mediating cell recruitment through interactions with other cells that, in turn, synthesize macrophage chemokines [40,41]. SEA is known to induce the expression of TNF [14,26,42], therefore, we reason that it might be only after granuloma initiation, at which point significant numbers of macrophages are in contact with the egg, that TNF is produced above the threshold to induce these chemokines. In addition, the close cell-to-cell contacts following the initiation of granuloma formation and epithelioid transformation may be vital; if TNF is acting in both an autocrine and paracrine manner, then the cell-to-cell interaction would allow for maximal signal exchange between cells, the optimal amplification of this signal and subsequent expression of chemokines [43,44]. Epithelioid transformation is primarily associated with Th2-polarized immune responses involving IL-4/IL-13, expression of which can occur in the context of innate immunity alone [42,45–47]. Therefore, it is not surprising to observe epithelioid transformation in the absence of TNF. Chronic mTORC1 signaling, which does not require adaptive immunity, can also induce epithelioid transformation [48].

We have recently shown that *S. mansoni* eggs, upon reaching maturity, induce granuloma formation that benefits the parasite by extruding the egg into the environment [4]. This would be achieved by mature egg stage-dependent secretion of antigens such as omega-1 [7,11]. Here, we show that recombinant omega-1 recruited macrophages rapidly, similar to SEA. Our finding supports the hypothesis that omega-1 is sufficient yet dispensable for early macrophage recruitment. This may have parallels in observations regarding its role in granuloma formation; omega-1 knockdown eggs form granulomas in the mouse, albeit smaller ones, suggesting other egg antigens such as IPSE could contribute to this process [16,17].

In addition, we have demonstrated that the omega-1 RNase activity is required for macrophage recruitment. This finding indicates that omega-1 does not act directly as a chemoattractant, and that recruitment must be mediated through downstream effects stemming from its RNase activity. Prior work has shown that its RNase activity mediates Th2 polarization through inhibition of protein synthesis in dendritic cells [21,22,30]. In the context of the 6-hour recruitment assay performed herein, we speculate that the protein is taken up by epithelial cells that line the hindbrain ventricle cavity, perturbing cellular homeostasis by an RNase-induced inhibition of protein synthesis and in turn, inducing cell stress signals which would trigger macrophage recruitment.

As for tuberculous granulomas [47,49], we expect this report will stimulate the use of this facile model to dissect mechanisms underlying the genesis of schistosome egg-induced granulomas, the main drivers of schistosomiasis pathogenesis and transmission.

## Supporting information

**S1 Fig. TNFR1 mutant zebrafish larvae have increased infection burden.** WT and TNFR1 mutant zebrafish larvae were systemically infected at 36 hours post-fertilization via caudal vein injection with 75 CFU *Mycobacterium marinum*, and then imaged at 4 days post-infection for bacterial burden. (A) The two animals closest to the mean. Scale bar, 300 μm. (B) Quantification of bacterial burden, with the two red data points corresponding to the animals in (A). Horizontal bar, means. Statistics, Student's t test. FPC: fluorescent pixel counts.
(TIF)

## Author Contributions

**Conceptualization:** Kevin K. Takaki, Lalita Ramakrishnan, Antonio J. Pagán.

**Formal analysis:** Kevin K. Takaki, Lalita Ramakrishnan.

**Funding acquisition:** Wannaporn Ittiprasert, Paul J. Brindley, Gabriel Rinaldi, Matthew Berriman, Lalita Ramakrishnan.

**Investigation:** Kevin K. Takaki.

**Methodology:** Kevin K. Takaki, Francisco J. Roca, Wannaporn Ittiprasert, Paul J. Brindley.

**Project administration:** Lalita Ramakrishnan.

**Resources:** Francisco J. Roca, Gabriele Schramm, Ruud H. P. Wilbers, Wannaporn Ittiprasert, Paul J. Brindley, Gabriel Rinaldi, Matthew Berriman.

**Supervision:** Lalita Ramakrishnan, Antonio J. Pagán.

**Validation:** Kevin K. Takaki, Lalita Ramakrishnan, Antonio J. Pagán.

**Visualization:** Kevin K. Takaki.

**Writing – original draft:** Kevin K. Takaki, Lalita Ramakrishnan, Antonio J. Pagán.

**Writing – review & editing:** Francisco J. Roca, Gabriele Schramm, Ruud H. P. Wilbers, Wannaporn Ittiprasert, Paul J. Brindley, Gabriel Rinaldi, Matthew Berriman, Antonio J. Pagán.

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
