## [Decision Letter · Decision Letter 0]

18 Oct 2020

Dear Ramakrishnan,

Thank you very much for submitting your manuscript "Tumor Necrosis Factor and Schistosoma mansoni egg antigen Omega-1 shape distinct aspects of the early egg-induced granulomatous response" for consideration at PLOS Neglected Tropical Diseases. As with all papers reviewed by the journal, your manuscript was reviewed by members of the editorial board and by several independent reviewers. The reviewers appreciated the attention to an important topic. Based on the reviews, we are likely to accept this manuscript for publication, providing that you modify the manuscript according to the review recommendations, particularly those of Reviewer 2.

Sincerely,

John Pius Dalton, PhD

Associate Editor

Sergio Costa Oliveira

Deputy Editor

Reviewer's Responses to Questions

**Key Review Criteria Required for Acceptance?**

**Methods**

-Are the objectives of the study clearly articulated with a clear testable hypothesis stated?

-Is the study design appropriate to address the stated objectives?

-Is the population clearly described and appropriate for the hypothesis being tested?

-Is the sample size sufficient to ensure adequate power to address the hypothesis being tested?

-Were correct statistical analysis used to support conclusions?

-Are there concerns about ethical or regulatory requirements being met?

Reviewer #1: (No Response)

Reviewer #2: This manuscript uses a new model system for schistosome research in zebrafish. The full model is described in a recently accepted paper, and this manuscript describes some additional studies of the role of TNF and Omega-1 in granulomatous responses. Although the two parts of the paper both have a good rationale, they do appear quite separate, and do not link together well as a single study.

**Results**

-Does the analysis presented match the analysis plan?

-Are the results clearly and completely presented?

-Are the figures (Tables, Images) of sufficient quality for clarity?

Reviewer #1: (No Response)

Reviewer #2: Results are presented well and described in a complete and clear manner.

**Conclusions**

-Are the conclusions supported by the data presented?

-Are the limitations of analysis clearly described?

-Do the authors discuss how these data can be helpful to advance our understanding of the topic under study?

-Is public health relevance addressed?

Reviewer #1: (No Response)

Reviewer #2: The paper is very well written, and for the most part conclusions are backed up by the data. One exception is the statement "we speculate that the protein is taken up by epithelial cells that line the hindbrain ventricle cavity, perturbing cellular homeostasis by an RNase-induced inhibition of protein synthesis and in turn, inducing cell stress signals which would trigger macrophage recruitment" - as indicated this is highly speculative and is not backed up by any data. It is a valid hypothesis and this paper would be much improved by producing some data to support this hypothesis.

**Editorial and Data Presentation Modifications?**

Reviewer #1: (No Response)

Reviewer #2: (No Response)

**Summary and General Comments**

Reviewer #1: This is a short but interesting paper, presenting the use of a novel animal platform to explore a protein released by schistosome eggs driving the formation of granuloma and more specifically macrophage recruitment.

There is a large amount of background in the Results section which needs to be moved to the introduction or discussion.

The reference to an accepted paper (Takaki et al., 2020)(Cell Host & Microbe, accepted) is problematic and impossible to review, while some efforts have been to demonstrated key points (fig s1). 

In Figure 2D in omega-1 samples the two outlier significantly increase the average of this condition, what do the authors think of these samples, since this is not seen in 2B? 

The importance of RNase activity in mac recruitment is clearly shown in DEPC treatment. 

The authors mention that a omega1 KO shipments wasn’t viable so they performed some other experiments, and then they had another shipment which is shown in 2e? This is not clear.

The authors comment about RNase activity with their samples, but what about cytotoxity against zebra fish cells specifically? This is important since O-1 is a major hepatotoxin. Does DEPC treat eliminated cytotoxicity also?

Reviewer #2: This study uses an important and novel new model to confirm some previous findings in mice. It contains some interesting data, however their novelty is limited by the fact that both findings (the role of TNF and Omega-1 in granuloma formation) have previously been tested in mice. Furthermore, the two parts of the manuscript did not lead naturally on from one another and it appeared to be two separate studies. 

Major comment:

Further work should be carried out to determine the mechanistic basis of RNAse-dependent macrophage recruitment to Omega-1. The authors propose a mechanism through epithelial cells - data should be provided to either back up this claim, or alternatively, Omega-1 could be acting directly on macrophages - the authors could address this by using WT or RNAse-deficient Omega-1 and assess whether either proteins induce isolated (human, mouse or fish) macrophage chemotaxis in vitro.

Minor comment:

Page 10: "The omega-1 deficient SEA retains ~20% of the omega-1 RNase activity (not shown), suggesting that even though reduced compared to wild type eggs, it is may still be sufficient for macrophage recruitment (Ittiprasert et al., 2019)." This statement should be clarified - does the knockdown only reduce Omega-1 expression by 80% or is there RNAse activity from some other source?

PLOS authors have the option to publish the peer review history of their article (what does this mean?). If published, this will include your full peer review and any attached files.

Reviewer #1: No

Reviewer #2: Yes: Dr Henry J McSorley
---

## [Decision Letter · Decision Letter 1]

17 Nov 2020

Dear Sergio Costa Oliveira,

We are pleased to inform you that your manuscript 'Tumor Necrosis Factor and Schistosoma mansoni egg antigen Omega-1 shape distinct aspects of the early egg-induced granulomatous response' has been provisionally accepted for publication in PLOS Neglected Tropical Diseases.

Best regards,

John Pius Dalton, PhD

Associate Editor

Sergio Costa Oliveira

Deputy Editor

Reviewer's Responses to Questions

**Key Review Criteria Required for Acceptance?**

**Methods**

-Are the objectives of the study clearly articulated with a clear testable hypothesis stated?

-Is the study design appropriate to address the stated objectives?

-Is the population clearly described and appropriate for the hypothesis being tested?

-Is the sample size sufficient to ensure adequate power to address the hypothesis being tested?

-Were correct statistical analysis used to support conclusions?

-Are there concerns about ethical or regulatory requirements being met?

Reviewer #2: (No Response)

**Results**

-Does the analysis presented match the analysis plan?

-Are the results clearly and completely presented?

-Are the figures (Tables, Images) of sufficient quality for clarity?

Reviewer #2: (No Response)

**Conclusions**

-Are the conclusions supported by the data presented?

-Are the limitations of analysis clearly described?

-Do the authors discuss how these data can be helpful to advance our understanding of the topic under study?

-Is public health relevance addressed?

Reviewer #2: (No Response)

**Editorial and Data Presentation Modifications?**

Reviewer #2: (No Response)

**Summary and General Comments**

Reviewer #2: (No Response)

PLOS authors have the option to publish the peer review history of their article (what does this mean?). If published, this will include your full peer review and any attached files.

Reviewer #2: **Yes: **Dr Henry J McSorley

---

## [Editor Report · Acceptance letter]

12 Jan 2021

Dear Professor Ramakrishnan,

We are delighted to inform you that your manuscript, "Tumor Necrosis Factor and *Schistosoma mansoni* egg antigen Omega-1 shape distinct aspects of the early egg-induced granulomatous response," has been formally accepted for publication in PLOS Neglected Tropical Diseases.

Best regards,

Shaden Kamhawi

co-Editor-in-Chief

Paul Brindley

co-Editor-in-Chief
